# The Cultural Cognitive Development of Personal Beliefs and Classroom Behaviours of Adult Language Instructors: A Qualitative Inquiry

**DOI:** 10.3390/brainsci8120220

**Published:** 2018-12-11

**Authors:** Luis Miguel Dos Santos

**Affiliations:** Woosong Language Institute, Woosong University, Daejeon 34514, Korea; luisdossantos@woosong.org; Tel.: +82-010-6887-0342

**Keywords:** adult teaching and learning, cultural influence, language teaching and learning, life-long learning, teachers’ professional development

## Abstract

The researcher employed personal belief system (PBS) theory as the theoretical foundation for this study because it holds that teachers’ PBSs may influence their teaching behaviours, teaching styles, and pedagogies in classroom practice due to cultural influences. The purpose of this qualitative study was to explore how teachers’ personal beliefs influence how they teach and how their approach may align with or diverge from cultural expectations in a private adult learning facility for English learning in Macau Special Administrative Region, China. The participants in this study were classroom teachers in a learning community who believe in collaborating to create environments for best practices. Two main research questions guided this study: (1) What is the relationship between teachers’ personal belief systems and their classroom practice; and (2) How does a teacher’s educational experience as a K-12 student affect their pedagogy in an adult English language learning program? Three types of data collection methods were employed: interview, classroom observation, and field note taking. The findings showed that teachers utilize their personal belief systems to engage their students through interactive teaching strategies, which was counter-intuitive for both teachers and students who had been taught with Eastern teaching styles. This study contributed to personal belief system theory and broadens the understanding of the perspectives and concepts of English teaching and supervision. The beliefs of teachers influenced their understanding of teaching and their classroom practices.

## 1. Introduction

English language teachers should understand how their current teaching beliefs and practices mesh with indigenous cultural approaches to teaching and learning in order to develop appropriate, culturally informed pedagogies for students [1,2]. Teachers’ examination of their personal beliefs is especially important in areas where Eastern and Western cultures and ideologies mix on a regular basis, such as in Macau, China. Macau is a Special Administrative Region governed by China and a major international tourist and hospitality-oriented city known primarily for resorts, gambling, fine dining, and shopping. However, English comprehensive skills are gradually declining in Macau [3] and many working professionals are unable to use the English language to communicate with visitors. A large number of middle-aged adults entering hospitality industries as career switchers have had little or no prior education in English. As a result, Macau has a large number of adult residents who are unable to communicate effectively in English with travellers [4,5,6].

In order to work effectively in the hospitality industry, many residents enrol in different types of English language learning programs [7,8,9]. Recent data from the government indicated that more than 72% of Macau residents are in the workforce [10,11], with a significant portion of them working in the hospitality industry. To assist these groups of adult learners, adult English language learning programs have been established, such as the Direcção dos Serviços de Educação e Juventude (Department of Education and Juvenile Services; DSEJ), private local educational learning centres, higher education institutions, and community centres. The purpose of this research is to explore how English language teachers’ personal beliefs influence their classroom practices at an adult education learning centre in Macau [12].

This study addresses four issues. First, English teachers’ personal beliefs about language teaching and learning may influence their current classroom practice. Teachers may have personal beliefs about their teaching philosophy, teaching goals, their understanding of their role as teachers, and previous learning experiences. Students’ learning in any learning environment is influenced by a comprehensive mix of variables. These variables may include how teachers instruct, or their pedagogical style. Both students and teachers entering the classroom have varying personalities, personal beliefs, personal preferences, and personal learning styles, which could positively or negatively influence the learning environment. Therefore, if teachers are able to share their sources of personal belief and receive feedback about language teaching and learning, their current classrooms will benefit from an improved learning environment.

Second, research about English teaching [13] has shown that it is helpful for teachers to express their understanding of classroom practices. This is because listening to peers feedback may encourage teachers to share their own teaching experiences. Such shared ideas may eventually be implemented by the teaching staff. Teachers are thus able to acquire empirical knowledge about English teaching and learning, which they can apply to their own classroom practices.

Third, Macau, where the researcher lives and works, is a Special Administrative Region governed by China. Macau is a major international tourist city known primarily for resorts, gambling, fine dining, and shopping. It is also a cultural centre, where Portuguese culture meets Chinese culture. Every year, Eastern and Western cultural attractions draw a large group of visitors who enjoy the historical heritage of the area [5]. However, there is a problematic trend in regard to Macau residents’ usage of the English language. English comprehension skills have gradually declined in the current decade. More importantly in an international hospitality-oriented city, many working professionals are unable to use the English language to communicate with visitors [3].

During the late 1900s, a larger number of immigrants entered colonial Portuguese Macau than in any other period in its history [5,6,14,15]. Currently, many of these immigrants are reaching middle age. Therefore, there are three major groups in Macau: native adult Macau residents, young Macau-born residents, and immigrants. The government has indicated that a large number of middle-aged adults entering hospitality industries as career switchers have had little or no prior education in English. As a result, Macau has a large number of adult residents who are unable to communicate with travellers in English.

As a large number of Macau residents cannot communicate in English and in order to gain promotions in the hospitality industry, many residents enrol in different types of English language learning programs. Recent data from the government indicated that more than 72% of Macau residents are in the workforce [10,11]. In addition, recent data from the government also indicated that only 1.9% of Macau residents are currently unemployed [10,11]. To assist these groups of adult learners and their family members in attaining a higher socioeconomic level, adult English language learning programs have been established. Currently, many types of organisations provide English language learning programs for adults, such as the Direcção dos Serviços de Educação e Juventude (Department of Education and Juvenile Service), known as the DSEJ, private local educational learning centres, higher education institutions, and community centres.

Last, there is little qualitative research on teachers’ personal beliefs about teaching the English language in Macau, particularly in the field of adult learning. Therefore, it is important to study the behaviours of teachers involved in adult English language learning programs. More importantly, this research provides the opportunity for teachers of an adult learning program to understand how their personal beliefs may influence their classroom practices.

### 1.1. Personal Belief System Theory

The personal belief system (PBS) theory developed by Kindsvatter et al. is used as a theoretical foundation for this study because it holds that teachers’ PBSs may influence their teaching behaviours, teaching styles, and pedagogies in classroom practice [2]. Belief is a sense or principle held by a group of people, as well as an opinion based on supported evidence, which impacts behaviours as a social phenomenon. Educators stress the need to encourage foreign language teachers to enhance their teaching and learning strategies. However, teachers state that there are grey areas in improving their current pedagogies. The most powerful influences on the enhancement of teaching and learning are instructors’ understanding of their own teaching practices and of the foundations of teaching and learning. Kindsvatter et al. described two aspects of PBS upon which behaviour is based: the intuitive and the rational. This theory posits that teachers’ personal beliefs and personal experiences influence their teaching behaviours, teaching styles, and pedagogies. These bases are further divided into two separate belief systems: unexamined beliefs and informed beliefs. Figure 1 below depicts Kindsvatter et al.’s personal belief system [2].

#### 1.1.1. Unexamined Beliefs

##### Experience-Based Impressions

Kindsvatter et al. argued that teachers’ service experiences usually direct what are deemed as being appropriate or useful exercises in their classrooms. Kindsvatter et al. suggested that, regardless of status, in-service teachers, pre-service teachers, junior teachers, and student teachers should examine personal beliefs that may influence their current or future teaching practices [2].

##### Traditional Practice

Kindsvatter et al. observed that traditional practices may differ between different cultures and settings. General practices and exercises used by large numbers of teachers and educational institutions tend to originate from traditional wisdom [2]. For instance, in a semester-based school, mid-December is always the end date of the fall semester. To give another example, when school principals step on the stage of a school auditorium, they typically deliver a long speech in front of everybody in the school. Therefore, teachers should have their own teaching styles, teaching curricula, and teaching pedagogies within their own cultures and school settings. When junior teachers and pre-service teachers enter a school, they can observe what veteran teachers do. They can also learn from veteran teachers’ experiences and combine them with their own understanding in order to create their own teaching pedagogies. Teachers in an adult foreign language learning program may believe that memorising and dictating vocabulary are the primary strategies in regard to studying foreign languages, based on their own experiences or traditional practices of studying language. Therefore, teachers may fail to develop their own teaching strategies and practices [16]. 

##### Personal Needs

Personal needs might include classroom management and teachers’ expectations of students. Kindsvatter et al. suggested that personal and mental health needs influence teachers’ perceptions and behaviours [2]. Researchers found that teachers like to arrange interactive activities with foreign language learners by pairing them with other learners. Teachers are expected to have good classroom management and expectations for their own class in order to achieve their teaching goals. For instance, in a library class, all students should be silent and return all assigned books to the correct place. In addition, in a physical education class, all students should be lined up in order to ensure the teacher has direct and orderly control.

#### 1.1.2. Informed Beliefs

##### Pedagogical Principles

Pedagogical principles are hard to define in classrooms. One teacher’s definitions are not the same as that of another. Some teachers believe punishment is a way of teaching, some believe assignments and homework are an effective method of teaching, and some believe interactions between peers constitute teaching. In fact, the pedagogical principles of English teachers could be shaped by their previous experiences as K-12 students. It is important for teachers to define what kinds of pedagogical principles to employ in order to develop activities for foreign language teaching [17]. However, regardless of type, teachers’ pedagogical principles are influenced by their own personal beliefs [2].

##### Constructivist Approaches

Kindsvatter et al. stated that the basis of the constructivist approach is treating students as meaning creators. Definitions and examples of constructivist approaches vary from teacher to teacher. However, for foreign language teachers, students’ meaning could come from their personal beliefs [2]. If a foreign language teacher believes that interaction is the way to learn a foreign language, it could be because of his or her positive experiences of learning a foreign language during K-16 schooling.

##### Teacher Effectiveness Practices

What are common techniques for foreign language teachers? Teachers may observe from other experienced teachers at their workplace, draw on previous learning experiences, and learn from books [13,17]. For foreign language teachers, techniques from their own teachers could be key to their own pedagogies. For example, if foreign language teachers engaged in interaction as a learning strategy during their K-12 experiences, they are more likely to use interaction as part of their own practice [2].

##### Research Findings

Kindsvatter et al. described research findings as discoveries through systematic methods of research into the content of education, instructional approaches, and learning effectiveness [2]. In general, experienced teachers usually read articles about teaching enhancements in order to open their minds to new ideas [2]. Junior foreign language teachers may not have enough teaching experience and time to access and integrate research findings into their own practices. Experienced teachers can discuss and share teaching and learning experiences with junior foreign language teachers so that they can discover effective practices for foreign language teaching. If the discussions and new teaching enhancements are meaningful and useful for the junior teachers, they may accept them and continue to change their practices. Because the discussions, enhancements, and experiences must be understood, they are defined as an informed beliefs system [18].

##### Scholarly Contributions

Kindsvatter et al. cited examples of teachers’ scholarly contributions to academia, including the creation of essays, models, theories, and judgments. However, it could be difficult for junior foreign language teachers to produce such high-level, informed knowledge in the field. At least five years of education-related experiences are recommended before doing so [2].

##### Examined Practice

Kindsvatter et al. suggest that examined practices serve as a foundation for teachers to try different teaching practices in order to provide the best practice to students. Foreign language teachers should try to exercise several new practices and reflect on them [2].

Foreign language teachers working in adult educational learning environments should understand their current teaching practices and previous learning experiences as learners in order to develop proper pedagogies for students. If all teachers understand their own beliefs, they can build comfortable learning environments that respect students’ needs. Students can also enjoy the benefits of new, meaningful, and suitable pedagogies in their foreign language learning experience [2].

### 1.2. Research Questions

The purpose of this research study is to explore the personal beliefs of five teachers at an adult English language-learning program in a private, local educational learning centre through the following research questions:1.What is the relationship between teachers’ personal belief systems and their classroom practice?2.How do teachers’ educational experiences as K-12 students affect their pedagogy in adult English language learning programs?

## 2. Methodology

A qualitative research design was used for this study. This design allowed the researcher to gain knowledge about the meaning of social phenomena, events, and English teachers’ understandings of their teaching practice in an adult learning centre in Macau. The researcher conducted two individual interviews with each participant, completed observations, and took field notes. A qualitative research design allowed the researcher to inductively analyse the data and build themes, answer the research questions, and discuss the findings, in order to understand teachers’ beliefs and their understanding of various teaching practices [19].

### 2.1. Participants

A convenience sampling strategy was used to locate English language teachers who were willing to share information about their educational experiences as students and how their experiences influenced their pedagogy and personal beliefs as teachers. The participants included five full-time employees with at least two years of English language teaching experience at an adult language learning centre in Macau. The sample was recruited through letters of invitation [19]. Detailed demographic information for the participants is shown in Table 1.

### 2.2. Data Collection

All participants took part in two individual interviews (pre-observation and post-observation). They were allowed to express their thoughts and feelings in either English or Cantonese. The pre-observation interview was held in a private room at the adult learning centre in Macau and lasted approximately 60 to 90 min. The interview questions focused on how participants recalled their experiences of learning English during the K-12 period and how they described and understood their current teaching experiences as English language teachers at a learning centre in Macau. The post-observation interviews were conducted after the researcher observed the participants in the classroom. This interview lasted approximately 60 to 90 min and was held in a private room at the adult learning centre in Macau [20,21,22].

Observation was employed as another form of data collection. During each observation, the researcher sat in the back corner of the classroom in order to avoid disturbing both the students and the teachers. The researcher observed one 60-min English class taught by each of the participants and used a field notebook to make notes during this time [23].

### 2.3. Data Analysis

A general inductive approach [24] to data analysis was used to analyse the collected data. The analysis process was guided by the objectives of the study whereby the researcher sought a better understanding of the participants’ educational experiences as students and how their experiences influenced their pedagogy and personal beliefs as teachers. Data analysis began after all interviews were professionally transcribed and member checked [19,24].

Following the procedure of Thomas [24], the transcripts were reread multiple times, as prescribed by the theoretical framework of interpretivism [25]. Preliminary themes began to emerge from the multiple readings of the transcripts. The researcher employed a constant comparison, as described by Merriam [19] to develop descriptive themes from the emerging trends in the data [20,21,22].

## 3. Findings and Analysis

The analysis of the interviews, observations, and field notes yielded six themes. These themes are identified in Table 2.

### 3.1. Student Purpose and Self-Direction

The first theme identified among the participants’ responses is related to the importance that teachers place on developing an understanding of student purpose and respecting their students’ sense of self-direction. The teachers maintained that adult learners came to school with a clear and definitive purpose in mind. As Danny explained, “Adult learners come to our centre. They must have direction; they must have a purpose. Usually, adults usually do not come here randomly. They come here for a purpose.”

In addition to possessing a purpose, an understanding of one’s purpose is an important component of the academic process. Amy noted, “They are all adults, they understand what they are doing here, and they have a purpose in coming to class.” For some students, this purpose was related to personal aspirational goals; for others, it was linked directly to their career objectives. For example, Betty said, “[Students] come here for a purpose. Some for the company, some for their own job.” This purpose serves as an important source of motivation and direction for adult learners. Teachers worked jointly with students to help them progress toward the achievement of their purpose. Three of the participants suggested that the primary responsibility of the teacher is to assist their students in achieving the purpose for which they are attending school. As Amy said, “The major influence of the current teaching experience is to see my previous students achieving their purpose.”

Most of the teachers argued that adult learners were self-directed and that learning was the student’s choice and responsibility. Because of this belief, teachers reported that they chose not to adopt a punitive style, and worked to establish a balance of power in the classroom. Betty explained this concept: “I am here to guide them in being successful, but I am not here to teach them. They are all adults, and some of them are even older than me. How much they learn is totally up to them.” Adult learners were viewed as capable and responsible students who should be entrusted to make their own educational choices.

Rather than ignoring the many external factors competing for the time and attention of adult learners, teachers took these responsibilities into account when working with this unique population of students. Adult learners were given leniency in regard to attending to their personal affairs and managing their time appropriately. Amy explained, “Adults come to school for knowledge and knowledge transfer. They are not children in secondary school. They understand what they need to do for work, for education, for their purpose, for their family, and so on.”

On the topic of student self-direction, Edwin stated, “This is all up to them. I am just a teacher; I can’t force them to learn.” Teachers took responsibility for presenting class materials and supporting student learning, but assigned students the responsibility of doing the work themselves.

### 3.2. Instruction Tailored to Students’ Needs

In providing instruction to adult English-language learners, teachers utilised a variety of educational methods. The second theme identified among the participants’ responses related to the teachers’ desire to make their instruction responsive to the needs of their students.

All of the teachers reported the belief that classroom instruction should be tailored to meet the students’ needs. Edwin explained this idea: “Teachers should always think about how to develop modules that match the needs of their students.” Danny reinforced this concept: “In this centre, I have learned about the idea of meeting students’ needs and helping them to achieve their targets.” In addition to meeting the individual needs of students, the teachers’ instruction was also designed with industry-based demands in mind. By making sure that lessons were in line with the current needs dictated by the industry, teachers were able to help students acquire job-appropriate skills and competencies.

Teachers aided student comprehension by varying the methods through which they presented educational materials. During the observation of Cathy’s classroom, a student expressed a lack of understanding of the brainstorming process. In order to help students successfully achieve their goals, Cathy gave examples by brainstorming on the whiteboard. Some teachers also utilised supplementary materials to serve as teaching aids in an effort to promote student comprehension. Danny designed some business and contemporary conversations in order to meet the needs of his students, such as telephone conversations and face-to-face interactive conversations with customers. Cathy strongly believed that teaching strategies should be flexible and contemporary: “I change my teaching materials almost bi-yearly in order to stay updated. News changes every day; if I continue to introduce old news or 10-year-old news, who wants to read it?” Creating clear and direct connections to the students’ real lives helped to generate interest in materials and enabled students to apply lessons beyond the classroom.

### 3.3. Teachers as Life-Long Learners

The third theme identified from the participants’ responses involved the ideology that the process of learning is life-long and that teachers should strive to continually expand their knowledge. The researcher identified two main topics from the data pertaining to the concept of teachers as life-long learners.

Most of the teachers expressed the belief that teachers should continually strive to learn and develop their skills. In discussing her interactions within the community of English language teachers, Cathy said, “If we can share educational and teaching experiences with each other, I think we can produce much better outcomes.” The collaborative effort advocated by all teachers ultimately contributed to the improvement and refinement of the instruction provided by the teachers.

In addition to learning from other teachers, the participating teachers also expressed a willingness to learn from their students. Students were viewed by the teachers as being valuable resources for knowledge and continued professional development. To address this point, Amy stated, “I do not think I am a teacher here in the adult learning classroom. I am a student as well. I can learn new knowledge from my students. We share, we discuss, and we engage together as a group.”

Several participants also maintained that teachers should strive to be open-minded and willing to change. Danny argued, “I think teachers should try their best to change, in order to provide better environments for students. But if a teacher continues to be obstinate, it is kind of unfair to the students as well.” The assertion was that teachers who refuse to adjust to the needs of their students ultimately provide a disservice to their students.

### 3.4. Importance of Caring and Support

The fourth theme identified among the participants’ responses focused on the importance of caring about and supporting students. Care was described as the showing of concern for students’ well-being and mastery of the material. Support was characterised as the demonstration of effort that goes above and beyond the mere presentation of material to ensure that students gain mastery of the lesson. In expressing this theme, the teachers discussed negative perceptions of their early educational experiences that were primarily due to a perceived lack of support from their teachers.

In discussing their K–12 education, most teachers indicated that their experiences had been somewhat negative. Cathy said she found that the grading system during her secondary school years contributed to a sense of “suffer(ing)”. Edwin stated, “Well, of course I did (have negative experiences). Some people are just, you know, kind of mean. I disliked English because of the teacher. The teacher really matters for sure.” Betty similarly said, “My secondary school, it was a nightmare. There was only memorisation. I did not have the space to express what I wanted to learn. Secondary school provided me with a bad example and made me determined to not exercise that kind of practice here.”

For other students, negative perceptions of their early educational experiences pertained mainly to workload and time. The teachers generally felt that the excessive amount of assigned work was made more difficult by the limited amount of time allotted for the courses. Danny shared that, in addition to feeling there was a shortage of time in school, “there was just too much homework.” He added, “I felt very unhappy.” Feelings of being rushed for time and having unreasonable workloads characterised most teachers’ memories of their early educational experiences.

The teachers reported that through their student experiences with their teachers, they learned the importance of being caring and supportive instructors. Betty recalled, “I had an English teacher who only want[ed] to get her pay check. She did not prepare for class. I mean, if she did not want to do her job, what is the point of coming to school and poisoning her students?” Betty maintained that the motivation for teaching should not be externally driven. Instead, teachers should be motivated by a genuine desire to help students learn.

The teachers expressed a desire to be better teachers than those they encountered during their formative years. Danny reported, “My secondary school experience (was) very negative. Currently, I am a teacher; I would like to provide a positive environment in which to help my students learn English.” Betty expressed a similar view, saying, “Sometimes, teachers can be very selfish. But I don’t want to be that kind of teacher.”

As a result of their experiences, several teachers reported that they strove to provide support to their students. As Amy mentioned in her pre-observation interview, she believed spending time with students was significant. Therefore, after her lessons, she spent around 30 min providing additional assistance to her students. Betty similarly explained, “Like my general practice, I always stay after school, even if it is a late night. I want to understand the needs of my students.” By making allowances to accommodate students and taking time to ensure that students were understanding the material, the teachers were able to demonstrate care and support for their students.

### 3.5. Dislike for Eastern Teaching Styles

Many distinctions exist between the teaching styles practiced by Western instructors and those practiced among Eastern teachers. The differences among these pedagogies extend to instructional styles, methods, and student-teacher interaction styles. The fifth theme to emerge from the data related to an overall dislike for the teaching style traditionally practiced by Eastern teachers. The researcher found three primary topic areas among the collected data that were related to the overall opposition to Eastern teaching styles.

All of the teachers discussed differences between Eastern and Western pedagogical practices and reported negative perceptions of Eastern teaching styles. Eastern teaching approaches tend to be characterised by a lecture format, in which the teacher presents information to passive, observant learners. Edwin argued that this style of instruction is not appropriate for adult learners and expressed a personal distaste for the style itself. He explained, “I have learned from university exchange experiences. I don’t want to use the traditional Asian teaching approach. I can see how Western education works and I dislike the traditional Chinese teaching approach.” Amy similarly noted, “Asian teachers like to use the top-down approach, (in) which students can only be the audience. Such a teaching approach is very negative. No voices can be expressed. Even if the teacher is wrong, the students can only listen.” The teachers argued that the top-down approach limits the ability of the students to contribute meaningfully to their educational experiences and removes their ability to challenge the instructor.

In response to their experiences with the top-down approach, many teachers reported or were observed modifying or foregoing the practice in favour of other teaching styles. Some teachers expressed an outright rejection of the top-down approach for their adult learners. Amy argued, “(My students) are in their adulthood. Some of them are even older than me. What is the point of using the top-down approach again?” By eschewing traditional Eastern teaching methods in favour of other methods, the teachers maintained that they were able to better serve the needs of their adult students.

### 3.6. Preference for Interactive Teaching Methods

The final theme the researcher identified was a general preference for teaching methods that involved close interaction between teachers and students. The participants described their use of instructional techniques that inspired the students to more fully interact with the lessons and with each other.

Most of the teachers reported a preference for interactive teaching methods over the top-down style typically practiced by Chinese teachers. The teachers indicated that they liked to incorporate role-playing and peer learning to increase student engagement. Danny argued, “Learning English should be interactive.” These activities can serve as an important motivational tool for students as well. Amy reported, “In terms of classroom motivation and behaviour, I try my best to create interactive activities in the classroom.”

## 4. Discussion

The purpose of this study was to capture the personal experiences and beliefs of five English language teachers participating in an adult English language-learning program for second language speakers in Macau, China. The research questions were designed to investigate how the teachers’ experiences and personal belief systems influenced their pedagogical choices.

### 4.1. Student Purpose and Self-Direction

Many of the jobs in Macau require a basic knowledge of the English language. Simple phrases and words can be learned quickly through interactions with customers, family, or friends; yet, there is still a very nascent understanding of the language. For most students, the purpose of learning English as a second language is to gain promotions in their industry, which is why they must enrol in English language learning programs. The teachers understood their students’ motivation to learn English, which coincides with Kindsvatter et al.’s [2] discussion of a teacher’s convergence of personal beliefs and the creation of a motivational environment in which to learn. The role of personal beliefs influenced the teachers’ actions regarding what types of teaching practices and strategies best served different learning groups [26]. The teachers understood that academic rigor in institutions may affect students’ abilities to learn, making teachers a propitious group to implement proper teaching practices and strategies within their classrooms [27,28,29]

Teachers were interested in their students’ underlying need to successfully complete their coursework in order to achieve their career and occupational goals. Kindsvatter et al.’s [2] research on teachers’ personal beliefs produced data that assist in comprehending student goals of matriculating and completing a course [30,31,32]. An analysis of the theory and results from this research revealed that teachers hold beliefs related to students’ purposes, especially when the students come from a different cultural and academic background. However, all of the participants agreed that when developing curriculum and teaching plans, the purpose of their students should be the prime consideration [33].

### 4.2. Tailored Instruction to Meet Student Needs

The teachers suggested that instruction should be tailored to meet the students’ career-related educational needs, making lessons more applicable to real-world situations. For adult students, workplace English indicates a business-oriented direction for teaching in an adult learning facility. It was found that teacher participants tailored their instruction to meet their students’ needs, which consequently matched their personal beliefs, coinciding with the second theme of this study [34]. According to Kindsvatter et al. [2], teachers’ intuitive belief systems inform their decision-making processes with experience-based impressions, which determines that their instruction be based on the most effective strategies, methods, techniques, and behaviours. The teachers expressed a need to change their teaching methods and tools to meet their students’ needs and developed individualised forms of instruction to meet those criteria [35].

Once the students’ purpose and motivations were understood, teachers described the interactive methods they employed to engage their students and meet their needs in regard to learning situational English [30,31,32]. It is important for teachers to understand their teaching strategies. When teachers do not have a clear notion about a new teaching strategy that is suggested by others, they will not apply it in their classrooms. Some teachers made adjustments to their curricula in order to present vocational English for a student who was teaching English in a school, and a combination of vocational and practical English for students in the hospitality industry [36]. These modifications were tailored to the students’ needs, thus allowing the teachers to create a motivational learning environment [2].

### 4.3. Teachers as Life-Long Learners

Life-long learning was an ideology that was expressed by all of the participants in this study. The participants expressed a sense of being supported within a learning community that understood learning as a continuous process. A community of learning reflects an external environment in which teachers are able to strive to become better teachers through peer collaboration and the sharing of best practices. The teachers understood that collaboration was a preferred method of continuous learning, which allowed them to be more successful in their delivery of concepts to their students. This finding is congruent with the rational components of teachers’ personal belief systems, including constructivist approaches, teacher effectiveness, research findings, scholarly contributions, and examined practice [2]. The teachers were involved in developing their pedagogy through continuous learning and collaboration with other teachers in order to be successful in meeting students’ needs. Teachers collaborated in regard to their experiences and insights within and outside the classroom [37]. One teacher described this perception of collaboration through her own personal beliefs, stating that, by sharing “education and teaching experience with each other, we can produce much better outcomes.”

A willingness to accept new information was important for teachers in regard to contributing to their experiences, as well as feeling that their experiences were being thoughtfully received and validated by their peers. Being open-minded about a student’s needs allows for individual attention and adaptation of pedagogy and curriculum to meet those needs. The decisions teachers make to adjust aspects of their teaching suggests that they understand that “effective decisions inevitably precede effective teaching” [2]. An effective decision-making process for these successful teachers was to remain open to the needs of their students and collaborate with other teachers in order to understand the best practices for their classroom and prevent past experiences in the classroom from obfuscating their judgment [30,31,32].

### 4.4. Importance of Caring and Support

As seen in the previously mentioned findings, teachers base their daily decision-making on their past experiences with formative education, contemporary pedagogical methods, and collaborative learning. The teachers reported that these combined experiences taught them the importance of being caring and supportive teachers, which involved being concerned about their students’ mastery of the coursework. Because “assumptions and beliefs are the basis for much of our everyday behaviour” [2], early educational experiences influenced how teachers approached their pedagogy and suggests why they had a constructivist approach to lesson planning and individualised instruction.

Participants were likely to stay for additional office hours to assist students with special needs. The participants’ behaviours corresponded to the constructivist approach found in Kindsvatter et al.’s theoretical framework. Students were regarded as meaning creators, seeking success and aiming to obtain mastery of their coursework. The teachers were facilitators, offering caring and supportive instruction to help their students meet these goals. All of the teachers believed that their learning experiences, care, and support from teachers were key to success [30,31,32].

### 4.5. Dislike for Eastern Teaching Styles

If teachers have negative beliefs about teaching practice, they are less likely to apply this practice in their classrooms [34,36,38]. The teachers in this study expressed a dislike for the top-down approach found in Eastern teaching styles, and which was directly experienced by the teachers when they were students. Students in traditional Eastern classrooms tend to be silent and listen to their teachers as an audience [2].

Echoing the above, one participant described assignments, homework, and classroom tasks as the critical tools for learning in traditional Asian classrooms. Because teachers understood their experiences and the negative associations directly linked to those experiences, they avoided excessive assignments that could decrease student motivation. Having these experiences also played an important, formative role in the development of pedagogy [2]. Many teachers described their preference for teacher-student interactions that allow students to express their opinions in the classroom and increase motivation. A rejection of the Eastern teaching styles that had formed negative impressions allowed the teachers to create a motivational learning environment to meet the students’ needs and purpose [30,31,32].

### 4.6. Preference for Interactive Teaching Methods

The participants of this study described a preference for interactive teaching methods in their classes, which are in direct contrast to the Eastern method of teaching, characterised by a lecture format and excessive classwork. In Eastern classroom environments, students are collectively engaged as a group and asked to share with each other; however, when they listen to their teachers, they tend to be obedient and sit as an audience. Researchers have concluded that the beliefs and experiences teachers bring into a classroom are potentially different from their students; therefore, they require some adjustment [39]. The teachers found that passive behaviour may be an obstacle to gaining knowledge through engagement in the classroom [30,31,32].

The behaviours of students were viewed as intuitively based; therefore, many of the participants engaged students in different types of learning activities, such as role-playing, in order to connect with their students in more appropriate learning [30,31,32]. The teachers improvised methods of delivery to increase classroom participation and engagement with the material [2].

Personal experiences and beliefs shaped the pedagogy of the participants in this study [2], which allowed them to embrace Western and reject Eastern styles of teaching. The final theme illustrates the application of a culmination of learning how to best instruct students through proven methods that engage and motivate them, so that students persist in class and succeed in obtaining personal goals [14,15,16].

## 5. Limitations

The present study focused on English teachers at a local private language learning centre for adult students, omitting in-service teachers in other types of teaching environments, as well as pre-service teachers and student teachers. The location in Macau, China was limited to a single school and a small sample of adult educators. Another limitation of this study was a lack of examination of the adult learning facility’s policy and the administrative powers that create such policies. Understanding the overarching guidelines and the teachers’ compliance with those guidelines may help to achieve a complete picture of the educational structure for an adult-only learning facility. Furthermore, these policies and procedures may differ from Western standards and examining them may result in a better understanding of the political environment that drives many educational policies.

## 6. Implications for Future Research

Information from this study may be helpful for the development of effective strategies for policymakers, teachers, school leadership, and teacher educators, who assess pedagogy and curriculum and implement teaching strategies directly into a classroom environment [18,29,40]. The findings can be extrapolated beyond the region in which the study took place and can help administrators at English language learning institutions and language learning centres in other regions of China. In addition, secondary school teachers and administrators may find this study to be beneficial because of the contribution it makes to personal belief systems. As many local teachers do not have experience in understanding personal beliefs and teacher educational development programs, the study’s findings could be beneficial for teachers, regardless of their teaching subjects.

Teachers might directly benefit from understanding the findings of this research because they can use them to understand the source of the connection between personal beliefs and classroom practices. From the findings of this study, teachers may gain a better understanding of their personal beliefs and teaching practice behaviours, which may encourage teachers and administrators to establish training programs that compare past teaching practice with more sensitive and current practices. Novice, pre-service, and student teachers may greatly benefit from understanding how their personal belief systems affect their classroom knowledge and strategies. The influence of peers and experienced teachers abrogates the novice teacher’s education in contemporary pedagogy, thereby affecting their ability to make effective decisions within the classroom that befit the situation. Furthermore, experienced teachers are better equipped with years of experience, training, and personal development, as opposed to novice teachers, because their decision-making is based on multiple perspectives and is not as easily swayed [15,33,38,41,42].

Professional development programs may benefit from the findings of this study through the development of approaches that engage teachers in thinking about the learning purposes of their adult students. Moreover, the programs could introduce strategies for engaging students in order to enhance their workplace knowledge of English through interactive teaching. The participants expressed an increase in student engagement from classroom activities, especially when those activities provided appropriate solutions to daily occupational needs. In addition, the participants recognised their students’ needs and tailored their classroom activities toward personal interactions with peers, which allowed them to practice their newfound skills with a foreign language [3,30].

## 7. Recommendation for Future Research and Conclusion

Upon completion of the preliminary analysis, PBS theory provided a lens through which to further interpret the data. Teachers’ belief systems may impact their teaching behaviours, teaching style, and pedagogy [2]. A PBS has the unique position of leading teachers toward an understanding of the large number of issues with and implications of their teaching. Beliefs and practices become congruent with each other [34]. This concept becomes more important when developing and implementing pedagogies, as well as improving daily teaching practices [43]. Conversely, teachers’ beliefs may not be reflected in their classroom practices. Because contextual factors, time constraints, and teaching activities were found to influence a teacher’s ability to adhere to personal beliefs and pedagogy, both congruence and incongruence in the classroom were factors that lent a better understanding of personal beliefs and classroom practices.

PBS theory is a two-tiered approach with a structural backbone of theoretical concepts. It includes two bases: the intuitive and the rational. Furthermore, each of these bases can be divided into two separate belief systems: unexamined beliefs and informed beliefs [2].

The findings of this study describe how teachers utilise their personal belief systems to engage their students with the classroom material through interactive teaching strategies, which was counter-intuitive for both teachers and students who had been taught with Eastern teaching styles. This research study contributes to personal belief system theory and broadens the extant understanding of the perspectives and concepts of English teaching and supervision. The beliefs of teachers influence their understanding about teaching, as well as their classroom practices [2], which are directly related to the findings of this study.

Future research could incorporate the observations of and interviews with students within the program for a better understanding of how they apply knowledge in their daily lives. Furthermore, the interviews could take place prior to the beginning of the session and at completion. Because the courses were not graded, unbiased opinions of students’ achievements may emerge and lend a better understanding of the efficacy of an interactive classroom environment. The interviews could explore teaching styles and personal attitudes toward the teachers. Additionally, expanding the sample through quantitative instruments may allow researchers to extrapolate these findings to a larger population or an entire teaching community. A combination or mixed methods study could juxtapose the perceptions of the teachers with the students, making for a more comprehensive study of the classroom environment in a private English language learning facility [8,9].

Future researchers could focus on pre-service, student, or novice teachers of the English language and their personal belief systems prior to engaging with other teachers and students in an English language adult learning facility. Having these nascent perceptions may grant insight into areas of teacher development courses within and outside universities. Researchers in other countries throughout Asia could replicate this study in other similar facilities and contribute to the findings of this study, which expands upon the current literature describing teachers’ personal belief systems in a classroom environment [8,9].

Teachers actively seek to understand the purpose and needs of their students enrolled in the class. In addition, teachers understand their roles as facilitators of knowledge acquisition. Because these students were adult learners, most were not enrolled to learn vocational English, but to acquire situational English that could be applied in their workplace. The findings of this study described teachers utilising their personal belief systems to engage their students with the material through interactive teaching strategies, which was counter-intuitive for both teachers and students who had been taught with Eastern teaching styles. This research study contributes to personal belief system theory and broadens the extant understanding of the perspectives and concepts of English teaching and supervision [8,9].

## Figures and Tables

**Figure 1 brainsci-08-00220-f001:**
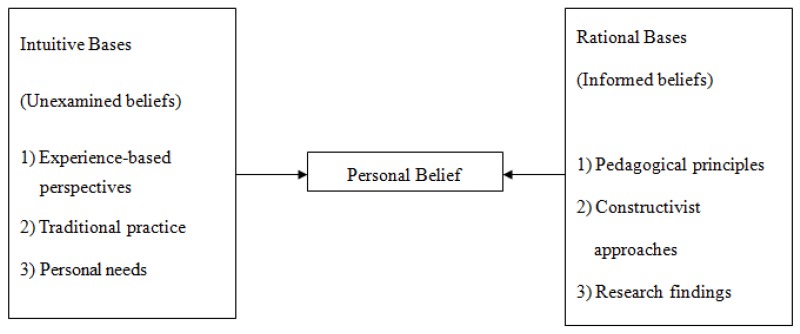
Personal belief system (PBS).

**Table 1 brainsci-08-00220-t001:** Demographic information for the participants. F, Female, M, Male.

Name	Gender	Grade or Subject of Teaching	Years of Experience in Teaching
Amy	F	Beginning, intermediate	8
Betty	F	Beginning, intermediate, advanced	10
Cathy	F	Advanced, vocational courses	5
Danny	M	Vocational courses, IELTS	9
Edwin	M	IELTS, vocational courses, beginning	8

**Table 2 brainsci-08-00220-t002:** Themes of the research findings.

Themes	Participants				
	Amy	Betty	Cathy	Danny	Edwin
(1) Student purpose and self-direction	Yes	Yes	Yes	Yes	Yes
(2) Tailoring instruction to meet student needs	Yes	Yes	Yes	Yes	Yes
(3) Teachers as life-long learners	Yes	Yes	Yes	Yes	Yes
(4) Importance of caring and support	Yes	Yes	Yes	Yes	Yes
(5) Dislike for Eastern teaching styles	Yes	Yes	Yes	Yes	Yes
(6) Preference for interactive teaching methods	Yes	Yes	Yes	Yes	Yes

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
