# Peer review of "The Cultural Cognitive Development of Personal Beliefs and Classroom Behaviours of Adult Language Instructors: A Qualitative Inquiry"

_brainsci, 2018, doi:10.3390/brainsci8120220_

Reviewer 1 Report

As the norm and more common papers are quantitative, it is better to add the term "qualitative study" or something similar in the title of a qualitative paper.

Not only names should be given to the teachers, years of experience, grade, and the gender should also be given. This can be added to the table.

The quotes are better off if they appear as an indenpendent paragraph. Smaller fonts maybe, but independent text, not merged with other text. 

In some sections of the paper, it seems like the participants are talking on behalf of students, or they are students. 

No paragraph should be composed of 2 sentences. 3 sentences are the minimum number of sentences needed to make a paragraph. 

Not much is discussed about the culture in the introduction, quotes, and the discussion. 

It is not clear if this study is to validate a theory, or is built on a theory. If based on a theory, the conclusion is not that the theory works. If it is conducted to test the theory, then the conclusion is that the theory is relevant in this context.

Please consult some well cited qualitative papers for the format. 

Author Response

Reviewer’s Feedback

 Reviwer#1

Q1: As the norm and more common papers are quantitative, it is better to add the term "qualitative study" or something similar in the title of a qualitative paper.

A: “A qualitative inquiry” has been added.

Q2: Not only names should be given to the teachers, years of experience, grade, and the gender should also be given. This can be added to the table.

A: A demographic information of the participants table has been added. Please refer to p. 7.

Q3: The quotes are better off if they appear as an indenpendent paragraph. Smaller fonts maybe, but independent text, not merged with other text. 

A: If the directed quotes and directed sharing exceed the 42 words limitation, I always create an independent paragraph. Otherwise, I keep it into the paragraph as the guideline indicated.

Q4: In some sections of the paper, it seems like the participants are talking on behalf of students, or they are students. 

A: This article captured the opinions and qualitative feedback from the participants, including the understanding and experience during their childhood and their current teaching. Therefore, some opinions could be their childhood’s experience and some of them are their experiences from their current teaching practice.

Q5: No paragraph should be composed of 2 sentences. 3 sentences are the minimum number of sentences needed to make a paragraph.

A: Acknowledged. Further information and details had been added, particularly for the theoretical framework section (p.4-6)

Q6: Not much is discussed about the culture in the introduction, quotes, and the discussion. 

It is not clear if this study is to validate a theory, or is built on a theory. If based on a theory, the conclusion is not that the theory works. If it is conducted to test the theory, then the conclusion is that the theory is relevant in this context. (Needed additional conclusion statement in the final section)

A: Acknowledged. Cultural information about the study and the city had been added. Information related to the theoretical framework had been added from section 6 and section 7.

Q7: Please consult some well cited qualitative papers for the format. 

A: Acknowledged.

The author has sent the article to an English editor for polishing.

Reviewer 2 Report

The current manuscript systematically examined the relationship between teachers’ personal beliefs and their classroom behavior, teaching styles and pedagogies. The author found that personal belief systems were employed by teachers to engage students with the material through interactive teaching strategies. This was counter-intuitive for both students and teachers taught with English learning styles. This important study enriches personal belief system theory and deepens our knowledge of English teaching and supervising. Overall, the study is well designed, rigorously performed, and experimental results carefully interpreted.

 My only concern is that the study was performed in Macau, China. Does the author consider the uniqueness of their English learning environment? How general are the findings? 

Author Response

Reviewer’s Feedback

 Reviwer#2

 Q: The current manuscript systematically examined the relationship between teachers’ personal beliefs and their classroom behavior, teaching styles and pedagogies. The author found that personal belief systems were employed by teachers to engage students with the material through interactive teaching strategies. This was counter-intuitive for both students and teachers taught with English learning styles. This important study enriches personal belief system theory and deepens our knowledge of English teaching and supervising. Overall, the study is well designed, rigorously performed, and experimental results carefully interpreted. My only concern is that the study was performed in Macau, China. Does the author consider the uniqueness of their English learning environment? How general are the findings? 

 A: Acknowledged. The researcher decided to investigate the unique situation in Macau as Macau and Hong Kong is the only two Special Administrative Region in the world. In order to address the unique situation in Macau, the researcher added several sections in the introduction section.

First, the researcher re-stated the English language and learning background in the first paragraph in the introduction section (p.2)

Second, the research outlined the four issues. The third point indicated the uniqueness of the Macau Special Administrative Region.

l  The uniqueness of Portuguese-Chinese cultural background in Macau. (p.2-3)

l  The uniqueness of the hospitality and tourism city setting in Macau (p.2-3)

l  The significant demands as a hospitality and tourism city of Macau (p.3)

Third, the research also provided a general understanding and approach for city-state with a similar situation, for the purpose of the application.

Fourth, the researcher had added some sections within the recommendation for future research and conclusion (p.16-17)

The author has sent the article to an English editor for polishing.

Reviewer 3 Report

Would like to see more discussion around the unexamined and informal beliefs--what are they and how

need to be clear in who the study will be looking at in the introduction not toward the middle of the paper

There are some places where citations are missing

Some of your sections are very limited, add more information

Author Response

Reviewer’s Feedback

Reviwer#3

Q1: Would like to see more discussion around the unexamined and informal beliefs--what are they and how need to be clear in who the study will be looking at in the introduction not toward the middle of the paper

A: Acknowledged. The researcher had added and expanded additional comments in the introduction section and theoretical framework section.

First, the researcher expanded the introduction section with the information about the unique situation in Macau. As below:

l  The uniqueness of Portuguese-Chinese cultural background in Macau. (p.2-3)

l  The uniqueness of the hospitality and tourism city setting in Macau (p.2-3)

l  The significant demands as a hospitality and tourism city of Macau (p.3)

l  The research provided a general understanding and approach for city-state with a similar situation, for the purpose of the application. (p.3)

Second, the researcher expanded the theoretical framework section with richer information (p.4-6). More information about the discussion of unexamined and informal beliefs was added.

Q2: There are some places where citations are missing

A: Acknowledged. The researcher changed the citation.

Q3: Some of your sections are very limited, add more information

A: Acknowledged. The researcher had expanded the introduction, theoretical framework (p.2-6), and the Recommendation for Future Research and Conclusion section (p.16)

The author has sent the article to an English editor for polishing.

Round  2

Reviewer 1 Report

The revision is satisfactory. The author disagreed with one of my comments, which I understand (explained why). So, the paper can be published as is.